# The Oral Microbial Ecosystem in Age-Related Xerostomia: A Critical Review

**DOI:** 10.3390/ijms252312815

**Published:** 2024-11-28

**Authors:** Xiao-Meng Pei, Lian-Xin Zhou, Man-Wah Tsang, William Chi-Shing Tai, Sze-Chuen Cesar Wong

**Affiliations:** Department of Applied Biology & Chemical Technology, The Hong Kong Polytechnic University, Hong Kong SAR 997700, Chinawilliam-cs.tai@polyu.edu.hk (W.C.-S.T.)

**Keywords:** aging, xerostomia, dry mouth, hyposalivation, oral microbiome, dysbiosis

## Abstract

Xerostomia is a widespread condition among the elderly, impacting as many as 50% of individuals within this demographic. This review aims to analyze the association between age-related xerostomia and the oral microbial ecosystem. Xerostomia not only induces discomfort but also heightens the susceptibility to oral diseases, including dental caries and infections. The oral microbial ecosystem, characterized by a dynamic equilibrium of microorganisms, is integral to the maintenance of oral health. Dysbiosis, defined as a microbial imbalance, can further aggravate oral health complications in those suffering from xerostomia. This review investigates the composition, diversity, and functionality of the oral microbiota in elderly individuals experiencing xerostomia, emphasizing the mechanisms underlying dysbiosis and its ramifications for both oral and systemic health. A comprehensive understanding of these interactions is vital for the formulation of effective management and prevention strategies aimed at enhancing the quality of life for older adults.

## 1. Introduction

Aging represents a multifaceted biological phenomenon that induces various physiological changes within the human body, including the oral cavity [1]. A notable oral health concern associated with aging is the development of xerostomia, which affects a significant portion of the elderly population [2]. Xerostomia is characterized by a subjective sensation of dryness in the oral cavity, frequently accompanied by a reduction in salivary flow and alterations in the composition of saliva [3,4,5,6]. Epidemiological studies indicate that the prevalence of xerostomia, or dry mouth, escalates with advancing age [7,8]. This condition can result in a range of discomforts, including difficulty in speaking, swallowing, chewing, and maintaining oral hygiene, thereby adversely impacting individuals’ quality of life. Furthermore, xerostomia has been linked to a heightened susceptibility to oral health issues, such as dental caries, oral infections, and immunological disorders [9], which can undermine overall oral health.

It is well known that the oral cavity serves as the second-largest reservoir of microorganisms within the human body, following the gastrointestinal tract [10]. The interplay between the microorganisms present in microbiomes and their environment is referred to as a microbial ecosystem, characterized by a dynamic equilibrium [11,12]. Saliva plays a crucial role in maintaining the delicate balance of the oral microbial ecosystem, providing essential functions such as mechanical clearance, antimicrobial defense, and the regulation of the oral microbiome [13]. Individuals with xerostomia, characterized by a progressive deterioration in salivary gland function and alterations in salivary composition, can significantly impact the homeostasis of the oral microbiome, leading to a state of dysbiosis. Certain oral microbial communities within this ecosystem are not only associated with oral health issues but are also linked to systemic diseases [14,15,16,17]. Consequently, this intricate and diverse oral microbial ecosystem is crucial for the promotion and maintenance of oral health, and any alterations within it may increase the risk of pathogenicity [11,13].

The frequency and severity of xerostomia symptoms are observed to increase gradually with age [18]. Xerostomia encompasses more than mere discomfort; it represents a complex pathological process involving the interaction of various factors. The sensation of dry mouth can be partially ascribed to reductions in salivary secretion, decreases in the thickness and elasticity of the oral mucosa, autoimmune changes, as well as the impacts of various diseases and pharmacological treatments in older adults [19,20,21]. These factors collectively affect the oral microbial ecosystem, which subsequently influences xerostomia, thereby perpetuating a detrimental cycle [22,23,24].

This review aims to conduct a comprehensive evaluation of the existing literature to enhance the understanding of alterations in the composition, diversity, and functionality of the oral microbiota in elderly individuals experiencing xerostomia. We will explore the putative mechanisms linking the dysbiosis of the oral microbial community to the pathogenesis of xerostomia. Understanding the alteration of oral microbiome in age-related xerostomia is crucial for the formulation of targeted interventions and management strategies aimed at addressing this widespread oral health issue in elderly people. This review is intended to be a significant resource for clinicians, researchers, and public health professionals dedicated to enhancing the quality of life and preserving the oral and overall health of the increasingly aging population.

We conducted a comprehensive literature search in PubMed and Google Scholar, focusing on English-language publications from 1970 to 2024. Our search strategy employed a combination of keywords related to the topic, including “xerostomia”, “dry mouth”, “aging”, “epidemiology”, “oral microbiome”, “diagnosis”, and “treatment”, etc., to refine the search and capture relevant studies. Additionally, we reviewed the reference lists of the articles identified to uncover further pertinent publications. The inclusion criteria encompassed original research articles, systematic reviews, and meta-analyses. Conversely, we excluded case reports, conference abstracts, editorials, non-peer-reviewed publications, and studies characterized by small sample sizes or methodological deficiencies that could undermine the reliability of the results.

## 2. Xerostomia

### 2.1. Definition and Classification of Xerostomia

Xerostomia, or dry mouth, is defined as the dryness of the oral cavity and can be caused by the reduction in salivary flow or the complete lack of saliva [9]. Xerostomia can be classified based on the underlying etiology into two distinct categories: true xerostomia, resulting from malfunction or impairment of the salivary glands, and symptomatic xerostomia, which is described as the subjective sensation of oral dryness, despite salivary glands maintaining a normal secretory function [25,26].

#### 2.1.1. True Xerostomia

True xerostomia, also referred to as xerostomia vera or primary xerostomia, is characterized by the dysfunction of the salivary glands, the specific etiological factors of which are outlined in Table 1 [4,26]. This particular type of xerostomia is the focus of this review. Several widely recognized factors contributing to xerostomia encompass physiological alterations, Sjögren's syndrome [27], and the adverse effects associated with pharmacological treatments [28]. This is particularly evident in individuals utilizing multiple medications [8] or those prescribed xerogenic agents [29]. Other contributing elements include dehydration, fluid imbalances, lifestyle choices, and radiation therapy targeting the head and neck region [19,30]. For example, in Sjögren’s syndrome, excessive infiltration of inflammatory cells, resulting in increased production of cytokines and degradation of tissue proteins, destroys the acinar cells and interferes with salivary synthesis, resulting in dysfunction of the salivary glands [31]. Multiple innate immune pathways, including the nuclear factor-κB pathway, are likely dysregulated in the salivary gland epithelium in Sjögren’s syndrome [32].

#### 2.1.2. Symptomatic Xerostomia

Symptomatic xerostomia, also called pseudo-xerostomia, xerostomia spuria, and symptomatica, is caused by non-salivary causes, which are shown in Table 1 [4,26,33,34]. Although this review does not specifically address symptomatic xerostomia, it is essential for clinicians and researchers to differentiate between true xerostomia and symptomatic xerostomia, as this distinction informs appropriate diagnostic and management strategies. In the case of true xerostomia, the emphasis should be placed on identifying and treating the underlying dysfunction of the salivary glands, which may necessitate the use of targeted therapies, saliva substitutes, or methods to stimulate salivary flow [35,36]. Conversely, pseudo-xerostomia may require a more comprehensive, multidisciplinary approach that integrates psychological, neurological, and behavioral interventions to address the subjective experience of oral dryness [37].

**Table 1 ijms-25-12815-t001:** Comparison of true xerostomia and symptomatic xerostomia.

	True Xerostomia	Symptomatic Xerostomia
Definition	Oral dryness induced by salivary gland malfunction	Subjective sensation of dry mouth with normal secretory function of salivary glands
Cause	Systemic disease (for examples, Sjögren’s syndrome, diabetes mellitus, sarcoidosis, herpes, hepatitis C, and end-stage renal disease) [33] Treatment (including medications and radiation therapy) side effects [26] Salivary gland problems Nutritional deficiencies Eating disorders (including anorexia and bulimia) [9]	Dehydration [9] Neurological dysfunction [38] Oral sensory dysfunction [39] Psychogenic causes Mouth breathing Idiopathic causes [40] Lifestyle (for examples, alcohol intake, coffee/tea intake, and smoking) [34]

### 2.2. Diagnosis of Xerostomia

The diagnosis of xerostomia requires a comprehensive approach that encompasses various assessments and diagnostic tools. As illustrated in Figure 1, the process of diagnosing xerostomia involves several aspects, including a thorough patient history inquiry [41], the use of questionnaires [42,43], physical examinations [44], measurements of salivary flow, blood tests [41], imaging tests [45], and biopsies [46]. In general, it is essential to closely examine the reported symptoms, medication usage, and relevant medical history [47]. Fox et al. developed a questionnaire designed to assess the severity of xerostomia, which may serve as an indicator of actual hyposalivation [48]. Moreover, a careful physical examination is fundamental to identify clinical signs pathognomonic for xerostomia, focusing on both extraoral and intraoral assessments. Several helpful signs have been proposed: (1) glandular hypertrophy, particularly of the parotid and submandibular glands, suggests a certain degree of salivary gland insufficiency [44]; (2) the assessment of minor salivary gland flow serves as a method for evaluating xerostomia [49]; (3) saliva present at the floor of the mouth is limited and appears foamy; (4) the tongue exhibits erythema, dryness, cracking, and a cobblestone texture [50].

Moreover, the objective measurement of salivary flow rates, both unstimulated and stimulated, is a crucial diagnostic tool. The average flow rates for normal unstimulated saliva and stimulated salivary flow are reported to be between 0.3 and 0.4 mL/min [51] and 1.5 and 2.0 mL/min [52,53], respectively. A flow rate of unstimulated whole saliva (UWS) below 0.1 mL/min and a stimulated whole saliva (SWS) flow rate below 0.7 mL/min, measured over a period of 5 to 15 min, are indicative of hyposalivation [54]. It is important to note that while stimulated saliva flow may fall within the normal range, the secretion of unstimulated saliva may still be inadequate. Therefore, it is essential to assess the flow rates of both types of saliva [50]. In certain instances, the confirmation of the underlying etiology of xerostomia may necessitate the utilization of blood tests, imaging scans, and a biopsy of the salivary gland or minor salivary glands.

The integration of a comprehensive diagnostic methodology, including subjective evaluations, objective measurements, and an extensive clinical examination, enables healthcare professionals to acquire an in-depth understanding of the patient’s condition. This approach facilitates the identification of the underlying etiology and the formulation of a tailored management plan that addresses the specific needs of individuals experiencing xerostomia.

## 3. Oral Microbiome Ecosystem

### 3.1. Oral Ecosystem and Oral Microbiomes

The oral cavity constitutes a dynamic and intricate ecosystem, inhabited by a wide variety of microorganisms. The distinct habitats within the mouth—including the teeth, gingival sulcus, tongue, buccal mucosa, hard and soft palates, and tonsils—along with the specific environmental conditions they provide collectively create a unique and complex oral ecosystem [12]. The oral ecosystem is composed of a diverse array of microorganisms, including bacteria, fungi, protozoa, archaea, and viruses, which are organized into biofilms. Collectively, these microorganisms are referred to as oral microflora, oral microbiota, or the oral microbiome [12,55,56], as illustrated in Figure 2. The precise number of microbial species within the oral microbiome remains an area of active investigation, with estimates subjected to variation. Among these microorganisms, oral bacteria have been identified as having the most significant influence on oral health [12]. The expanded Human Oral Microbiome Database (eHOMD) [57] currently lists 774 bacterial species, of which 58% have been identified and 74% successfully cultivated. The predominant phyla found in a healthy oral environment include *Firmicutes*, *Proteobacteria*, *Actinobacteria*, *Bacteroides*, *Fusobacteria*, and *Spirochaetes* [58]. Notably, 11 out of the 15 phyla present in the oral cavity engage in a symbiotic relationship with the host, conferring benefits without causing harm; these are classified as commensal bacteria [59,60,61].

Microorganisms in the oral cavity comprise not only resident species that have established colonization but also transient species that originate from other parts of the body and may be identified within the oral environment [62]. *Bifidobacteria*, which are Gram-positive prokaryotes, predominantly inhabit the human gastrointestinal tract. Among the more than thirty species within this group, only a limited number—specifically three species—are known to colonize the oral cavity: *B. denticolens*, *B. dentium*, and *B. inopinatum* [63,64,65]. Microorganisms belonging to the genus *Bifidobacterium* may enter the oral cavity via saliva and food remnants, where they typically persist for a limited duration [66,67]. Consequently, this type of microbiome is classified as transient flora within the oral cavity [12]. The presence of elevated levels of *Bifidobacteria* can disrupt the microecological balance of the oral cavity, potentially leading to the development of dental caries [65], despite their inclusion as active ingredients in oral health products, food items, beverages, etc.

The diverse microbial flora present in the oral cavity engage in complex interactions, making it challenging to definitively attribute the increase or decrease in a specific microorganism to the onset of particular oral diseases. A decrease in resident species that are typically present in low abundance (<1%) to higher levels (>1%) suggests a dysbiotic state within the oral ecosystem [68,69]. Under certain conditions, the normal flora of the oral cavity can exhibit pathogenic characteristics. Oral commensal bacteria may acquire genetic elements that endow them with new pathogenic capabilities, such as toxin production or evasion of the host immune response [70,71,72]. Table 2 summarizes the common beneficial, pathogenic, and opportunistic bacteria found in the oral cavity. Notably, *Haemophilus parainfluenzae* is recognized as a commensal bacterium within this environment [23,24,73]. However, research concerning the microbial flora associated with xerostomia remains insufficient and warrants further investigation.

### 3.2. The Importance of Oral Microbial Ecosystem

The microbial ecosystem within the oral cavity plays a crucial role in maintaining oral health. The microbiota establishes a biofilm, a complex three-dimensional structure [93,94,95], that adheres to various surfaces within the mouth [72,96]. This biofilm formation is a key factor contributing to the persistence of microorganisms in the oral cavity [59], thereby supporting oral homeostasis. The oral environment is characterized as a dynamic system, shaped by intrinsic biological host factors, individual behaviors, and external influences, including circadian rhythms that affect salivary flow and composition [97]. The human oral microbiome comprises a shared core microbiome alongside a unique variable microbiome, which is influenced by lifestyle choices and physiological differences, as well as factors such as oral health status, age, and sex that determine the diversity and abundance of oral microbiota [98]. For example, individuals adhering to a Mediterranean diet exhibited significantly higher levels of *Subflava* and *Prevotella* in their oral cavities compared to vegans, likely attributable to different metabolic patterns associated with these dietary choices [99]. Moreover, both patients with periodontitis and those with dental caries demonstrated elevated levels of *Streptococcus mutans* in their oral microbiomes [100,101,102]. In contrast, lower levels of taxa such as *Porphyromonas endodontalis*, *Alloprevotella tannerae*, *Filifactor alocis*, *Treponema*, *Lautropia mirabilis*, and *Pseudopropionibacterium* sp.*_HMT_194* [103] were identified in the saliva of elderly individuals, alongside a high β diversity index [104]. Additionally, significant differences in oral microflora were observed between genders, with females exhibiting a greater abundance of *Streptococcus* in their oral cavities [105].

Similar to the intestinal flora, various oral microbiota are acknowledged for their contributions to the ecological equilibrium of the oral cavity. Certain oral probiotics, such as *Lactobacillus*, have been shown to inhibit the proliferation of pathogenic bacteria, including *Aggregatibacter actinomycetemcomitans*, *Porphyromonas gingivalis*, *Prevotella intermedia*, and *Streptococcus mutans*, while exhibiting no effect on the growth of *Candida albicans* [106]. Another probiotic, *Streptococcus salivarius*, which adheres to the oral mucosa, synthesizes lantibiotic bacteriocins that mitigate the presence of pro-inflammatory molecules within the oral environment [107,108,109]. Beyond the influence of individual microbiomes, the oral microbiome may also exert systemic effects; for instance, oral nitrate-reducing bacteria are involved in the human nitrogen cycle, facilitating the production of nitrate and the release of nitric oxide, which plays a role in the regulation of blood pressure [110,111,112,113].

The phenomenon of colonization resistance exists in the oral cavity, whereby the oral microbiota inhibits the colonization of other pathogens [114]. Disruption of this stabilizing microbiological structure can lead to the emergence of various oral diseases, characterized by alterations in the composition of oral microbiomes. For example, an increased ratio of Gram-negative rods to spirochaetes has been associated with the development of gingivitis, while an elevation in *Prevotella melaninogenica* may contribute to the pathogenesis of oral lichen planus [115,116]. Furthermore, dysregulation of oral microbiomes has been correlated with systemic health issues; specifically, diabetes has been demonstrated to alter the bacterial composition in the oral cavity, resulting in a greater prevalence of pathogenic bacteria [117].

The microbial ecosystems present within the human body, particularly in the oral cavity, function as active contributors to health maintenance and disease prevention. A comprehensive understanding of the human microbiome is crucial for the advancement of both medical and dental sciences, significantly enhancing the health of individuals and populations. Next, we continue to unravel the microbiome profile in age-related xerostomia. This exploration will also provide insights into the potential targeted interventions, such as the application of probiotics, antimicrobial agents, or personalized oral care strategies, aimed at restoring the equilibrium of the oral microbiome and alleviating the adverse impacts of xerostomia.

## 4. Age-Related Xerostomia

Xerostomia, commonly referred to as dry mouth, is frequently observed in the elderly population and is generally linked to a decline in salivary gland functionality [118]. This decline in glandular activity leads to diminished saliva production, which in turn contributes to sensations of dryness, discomfort, and a range of other oral health complications. Saliva, a complex biological fluid synthesized by the salivary glands, is essential for the mastication process, as it aids in the formation of a cohesive, smooth, and easily swallowable bolus [119]. The reduction in salivary flow can significantly affect the intricate equilibrium of the oral microbiome. These alterations in the microbial composition may subsequently lead to the emergence of various oral health issues, such as dental caries, periodontal disease [120], and opportunistic infections [121]. Moreover, individuals suffering from chronic xerostomia may experience poorly fitting dentures, and alterations in taste perception. The presence of xerostomia can significantly diminish quality of life, adversely affecting speech, eating, and swallowing functions. Comprehending the intricate relationship between age-related xerostomia and the oral microbiome is critically significant, as it can guide the formulation of targeted management strategies aimed at preserving oral health and enhancing the quality of life for the elderly population.

### 4.1. Epidemiology of Xerostomia in Elderly People

The prevalence of xerostomia is recognized to escalate significantly with advancing age [8,28]. Numerous epidemiological studies have consistently demonstrated a higher prevalence of this condition within the elderly population relative to younger groups. In Mexico, 68.3% of older individuals (aged 60 years and above) experienced xerostomia and/or hyposalivation [122]. A longitudinal study conducted in two Swedish counties revealed that xerostomia was significantly more prevalent in women than in men across all age groups. At the age of 80 years, “often mouth dryness at night” was reported by 24.3% of women and 16.2% of men. Notably, the persistence of xerostomia was observed in 61.4–77.5% of the participants, while progression of the condition was reported by 11.5–33.0% and remission by 5.7–11.3% [7]. A systematic review and meta-analysis suggested that the overall prevalence of hyposalivation in older adults aged 60 years and above is 33.37%, based on a total of 3885 individuals across thirteen studies [123]. This prevalence is significantly higher than that found in younger individuals, specifically those under the age of 35 years, where it is generally approximately 10% [42,124]. Moreover, xerostomia is reported more commonly in women [125], smokers, and individuals with symptoms of depression [29]. Females were predominant in the moderate (72.2%) and severe grade (88.5%) xerostomia group [124].

The endeavor to generalize the prevalence of dry mouth presents significant challenges, primarily stemming from inconsistencies in measurement methodologies, case definitions, and sample characteristics across various studies, rather than a deficiency in epidemiological research. It is crucial to differentiate between studies that examine xerostomia and those that investigate salivary gland hypofunction (SGH). As a result, the reported prevalence estimates exhibit considerable variability; for instance, studies utilizing representative samples of older populations report xerostomia prevalence rates between 12% and 39%, with a weighted average of approximately 21%. In contrast, prevalence estimates for SGH among older adults demonstrate an even broader range, from 5% to 47%, underscoring the influence of variations in methodologies and case definitions employed in these investigations [126]. Despite this variability, the collective findings suggest that dry mouth is a common condition among older individuals, occurring with greater frequency than in younger populations. Notably, the majority of epidemiological studies on this condition have concentrated on older adult samples, with only two studies addressing younger adults [8,127], which reported a prevalence of approximately half that of older cohorts. Although these inconsistencies hinder direct comparisons of epidemiological estimates from studies involving older populations, it is reasonable to infer that approximately one in five older individuals experiences dry mouth [128].

### 4.2. Causes of Xerostomia in Elderly People

The onset of xerostomia among the elderly population is a multifaceted issue influenced by various factors that lead to diminished salivary gland function and decreased saliva production. It is crucial to identify and address these diverse etiological elements in order to implement effective management strategies for xerostomia associated with aging. It is widely accepted that salivary secretion decreases with age. Research indicates that both unstimulated and stimulated salivation in humans diminish with advancing age [129]. The primary factor contributing to xerostomia in the elderly population is the natural decline in the functionality of salivary glands associated with the aging process. As individuals age, their salivary glands may experience structural and functional modifications, such as atrophy of acinar cells, fibrosis, and reduced blood supply [130,131]. Histological analyses have demonstrated that the average volume of acini decreases by approximately 30% in the submandibular (SM) glands, in the sublingual (SL) glands by nearly 25%, and in the parotid (PAR) glands by around 12% [132], and there is a replacement of parenchyma by fibrous and adipose tissue [133] with age. Furthermore, there is a noted increase in the number of terminal deoxynucleotidyl transferase dUTP nick end labeling (TUNEL)-positive apoptotic cells within the SM glands with advancing age, indicating that cellular turnover and associated changes contribute to the dysfunction of salivary glands as individuals age [134]. Furthermore, animal studies have demonstrated a 60% decline in protein synthesis in older age [135]. These age-related alterations lead to a gradual decline in both the production and secretion of saliva, as well as modifications in the concentration and/or activity of its organic components. Longitudinal studies have consistently demonstrated a correlation between advancing age and the increasing prevalence of xerostomia, thereby highlighting the significant impact of physiological aging on this condition [136,137].

### 4.3. Oral Microbiota in Age-Related Xerostomia

The oral microbiome constitutes a significant component of the human microbiome, characterized by a variety of distinct niches within the oral cavity that host diverse microbial communities. This environment is populated by an array of microorganisms, including bacteria, fungi, viruses, archaea, and protozoa, which collectively form a complex ecological network that plays a crucial role in both oral and systemic health. Notably, the most common oral diseases, such as dental caries and periodontal diseases, are closely associated with microbiota. In light of the growing understanding of the oral microbiome, several innovative approaches aimed at modulating the microbiome to preserve and restore a healthy oral ecosystem have been developed.

The reduction in salivary flow associated with age-related xerostomia can have a profound impact on the delicate balance of the oral microbiome. Saliva plays a crucial role in maintaining a healthy and diverse microbial community within the oral cavity. It provides essential nutrients, facilitates the clearance of food debris and microorganisms, and contains antimicrobial factors that help regulate the growth and composition of the resident microbes [138]. However, when salivary production is diminished, as observed in the condition known as xerostomia, the oral environment becomes susceptible to significant microbial shifts. This dysbiosis, or imbalance, of the oral microbiome can lead to the overgrowth of harmful bacteria and a decline in beneficial microorganisms [13,139]. The proposed entire process is illustrated in Figure 3.

It is posited that diminished salivary flow is the principal factor influencing the alteration of the bacterial community’s composition. This assertion has been substantiated in individuals diagnosed with primary Sjögren’s syndrome, where salivary flow accounted for 90% of the observed variation among samples, in contrast to only 5% of the variation being attributable to the disease status [140]. A significant aspect of the alterations in bacterial composition associated with xerostomia is the colonization by nonoral bacteria, including coliforms and *Staphylococcus aureus*. Furthermore, there is a notable rise in both the prevalence of *Candida* species and the incidence of *Candida* infections [141]. The colonization of nonoral bacteria is likely a consequence of diminished immune function delivery, which is typically facilitated by saliva. Nevertheless, the limited community profiling studies conducted to compare the bacterial composition of oral communities in individuals with hyposalivation against control groups have yielded either no significant differences [142], or only minor variations, along with some conflicting results. For instance, two studies [143,144] reported an increase in the proportion of *Streptococci* within the tongue microbiota of patients diagnosed with Sjögren’s syndrome, while another study indicated a decrease in the salivary microbiota of the same patient population [140].

The current body of literature addressing the alterations in oral microbiota associated with age-related xerostomia is relatively limited. Most participants in these studies have experienced dry mouth as a result of pre-existing primary conditions. Nevertheless, the findings provide significant insights into the changes in oral flora that occur due to the different primary conditions (Table 3). Specifically, the oral microbiota of patients with xerostomia following radioiodine therapy (RAI) for differentiated thyroid carcinoma (DTC) exhibited a higher abundance of *Porphyromonas*, *Fusobacterium*, and *Treponema_2*. Subsequent analyses suggested that the *Porphyromonas* genus may serve as a significant community driver in the development of xerostomia. These alterations in microbiota composition, along with corresponding functional changes, may foster a pro-inflammatory environment. The identified dysbiosis in the oral microbiota, coupled with the dysregulation of inflammatory and antioxidant metabolic pathways, may further exacerbate the progression of xerostomia [145]. Additionally, a particular study has documented salivary dysbiosis in patients with primary Sjögren’s syndrome (pSS), emphasizing the reduction in *H. parainfluenzae* as a significant clinical feature within this cohort. Furthermore, it was found that *H. parainfluenzae* enhanced PD-L1 expression in A253 cells, and A253 cells pretreated with *H. parainfluenzae* exhibited a suppression of CD4 T cell proliferation in vitro [23]. Interestingly, Weng et al. identified that the primary phylum differences between xerostomia and healthy groups included *Actinobacteria*, *Firmicutes*, *Fusobacteria*, and *Proteobacteria*. A similar phylum shift was also observed between Sjögren’s syndrome and non-Sjögren’s syndrome populations [146]. Previous investigations have not successfully elucidated the specific bacterial flora changes in individuals with dry mouth, their interactions with the host, and the subsequent implications for oral health. This gap in knowledge arises from the variability in study populations, which differ in primary causes of dry mouth, age, and sample sizes. Currently, there is a notable deficiency in information regarding the impact of aging on oral flora, indicating a need for further research and validation in this area. A comprehensive understanding of the oral microbiota characteristics associated with age-related xerostomia is crucial for the development of therapeutic approaches to manage this condition.

Age itself serves as a significant risk factor for xerostomia, independent of underlying diseases or pharmacological influences [155,156]. Old age, age 65 and above, when adjusted for edentulism, current tobacco use, periodontal disease, caries level, medication count, and gender revealed six bacteria linked to this condition. These six taxa, at lower levels in the old age group, are found on or near tooth surfaces above or below the gum line. Three of the taxa, *Filofactor alocis*, *Porphyromonas endodontalis*, and *Treponema*, are known to be enriched with periodontal disease, and one, *Lautropia mirabilis*, is enriched with periodontal health [157,158,159,160]. In a recent comprehensive study of old age effects on the subgingival microbiome of US woman [161], levels of two out of four taxa, measured in both studies, *Porphyromonas endontalis* and *Lautropia mirabilis*, were lower in the >70-year-old versus the 50–59-year-old group, while they saw no difference in *Filifactor alocis* and *Alloprevotella tannerae* at that site. A recent study has identified potential age- or health index-related bacterial genera, including *Fusobacterium*, *Parvimonas*, *Porphyromonas*, *Aminobacter*, *Collinsella*, *Clostridium*, and *Acinetobacter* in the Chinese elderly population aged 65 years and older [162]. In addition, the abundance of *Akkermansia* and *Acholeplasma* were significantly decreased with the progression of aging, which may be related to the development of inflammation [162]. Nevertheless, the impact of age-related alterations in the oral microbiome on xerostomia is not yet fully understood.

One may speculate on a number of changes that occur in the mouths of the post-65 group. This can include endogenous physiological changes of aging, such as mucosal changes, changes in immune function, and changes in salivary flow. Interestingly, comparison to studies of xerostomia patients using stimulated saliva for sample collection showed higher levels of *Treponema* and *Porphyromonas endodontalis* with low salivary secretion [14], the opposite of what is seen in the old age group [103], suggesting that the factors of age and reduced salivary flow may exert distinct influences on the composition of oral microbiota.

The aging process adversely affects the salivary glands, resulting in alterations to both the quantity (flow rate) and quality (e.g., ionic and protein composition, rheological properties, tribological characteristics) of saliva [163]. The field of geroscience has delineated and classified the cellular and molecular hallmarks of aging [164], which encompass genomic instability, telomere shortening, epigenetic modifications, loss of proteostasis, dysregulated nutrient sensing, mitochondrial dysfunction, cellular senescence, stem cell exhaustion, and altered intercellular communication. These hallmarks of aging may also have implications for oral health [155]. As saliva production declines with age, it raises the question of how the oral microbiome is affected. For example, while *Candida* species are typically part of the normal oral flora, conditions such as immunodeficiencies, malnutrition, or xerostomia can lead to the risk of overgrowth in the elderly and subsequent infection [165]. However, there is currently a paucity of studies examining the effects of aging on oral flora. This raises several pertinent questions: What is the relationship between aging and xerostomia? Do alterations in oral flora lead to subsequent changes in immune response and inflammation? Could these changes be implicated in the reduction in saliva production, xerostomia, Sjögren’s syndrome, and other related conditions in the elderly? Drawing from published research on the alterations in the gut microbiome during aging in rodent models [166,167,168], it is reasonable to hypothesize that significant changes in the oral microbiome will also occur in aging mice. Therefore, investigating the relationship between oral microbiota and age is a valuable endeavor.

## 5. Treatment for Xerostomia

Current approaches for managing xerostomia (Table 4) primarily aim at symptomatic relief and/or restoration of normal salivary secretion. In practice, these treatment options are employed based on the etiology and severity of xerostomia and SGH [169,170,171]. In treatment decision-making, if the patients retain residual salivary function, restoration of normal salivary secretion represents a more preferable choice than symptomatic treatment [170]. Nowadays, the prevailing treatments for xerostomia involve the use of salivary stimulants (sialagogues) and saliva substitutes. Furthermore, several non-pharmacological interventions, including acupuncture, electrostimulation, and low-level laser therapy, as well as gene therapy, have emerged as strategies for xerostomia treatment.

### 5.1. Use of Salivary Stimulants and Saliva Substitutes

At present, salivary stimulants and saliva substitutes are popularly prescribed for managing xerostomia. Salivary stimulants, such as chewing gum, malic and critic acids, and parasympathomimetic therapeutics, are substances that can stimulate the production of normal saliva, thus improving xerostomia and complications in salivary gland hypofunction, whereas saliva substitutes (artificial saliva) are preparations that resemble natural saliva in terms of composition and biophysical properties, applied for lubricating and moisturizing the oral cavity to relieve the symptoms of dry mouth.

Salivary stimulants have been commonly used for improving salivary secretion in people with residual salivary gland function. Among various types of salivary stimulants, two parasympathomimetic drugs, pilocarpine and cevimeline, are FDA-approved for managing xerostomia, in which pilocarpine is for treating xerostomia caused by Sjögren’s syndrome and radiotherapy, whereas cevimeline is only for Sjögren’s syndrome-induced xerostomia [172,173]. They are cholinergic agonists that mediate hypersalivation by activating the muscarinic receptors within the salivary glands. Both are effective in improving xerostomia [172,174,175,176,177], but their use is associated with a number of side effects, such as excessive sweating, nausea, flushing, respiratory distress, palpitations, gastrointestinal tract disturbance, diarrhea, and frequent urination [173,178,179]. In addition, they are also contraindicated in patients with other comorbidities, including uncontrolled asthma, acute iritis, and narrow-closure glaucoma [173]. These adverse events limit their clinical use in xerostomia treatment.

Saliva substitutes have been widely applied in the palliative management of symptoms, particularly for those who are intolerant to pharmacological sialagogues or suffer from permanent damage to the salivary glands. Numerous saliva substitutes, available in various forms, such as spray, mouth rinse, and gel, have been launched [180]. They are formulated with thickeners, such as carboxymethyl cellulose, mucin, and xanthan gum, to imitate the rheological and lubricating properties of saliva, allowing them to provide oral lubrication and maintain oral mucosal moisture [181,182]. Furthermore, to mimic the antibacterial properties of natural saliva, some saliva substitutes are added with antimicrobial agents, such as lysozyme, lactoferrin, and lactoperoxidase, to inactivate the growth of microorganisms that cause dental caries and gingivitis [183]. The antibacterial activity of lysozyme is related to its hydrolytic action on the bacterial cell wall, which makes the bacteria susceptible to osmotic lysis [184]. Lactoferrin, an iron-binding glycoprotein, kills bacteria by depriving cariogenic bacteria of iron, which is essential for bacterial cell growth [185]. Lactoperoxidase catalyzes the oxidation of thiocyanate ions, generating hypothiocyanite ions that can impair the function of microbial proteins via the oxidization of their thiol groups, eventually causing bacterial cell death [186]. Antibacterial effects of the saliva substitutes containing these agents have been demonstrated in several in vitro studies [183,187,188]. For instance, Tonguc Altin et al. illustrated that salivary substitutes supplemented with lysozyme or lactoferrin exhibited significant inhibitory effects on *Streptococcus mutans* [183]. Nevertheless, some studies revealed that commercially available saliva substitutes comprising these antimicrobial agents are ineffective in controlling the oral cariogenic microflora [181,189,190,191]. In addition to thickeners and antimicrobial agents, some saliva substitutes also include minerals, such as phosphates, calcium, and fluoride, to impose a protective effect on the teeth and gums [192]. While saliva substitutes generally improve xerostomia, they only provide a short duration of symptomatic relief; therefore, a repetitive administration of these agents is essential.

### 5.2. Non-Pharmacological Stimulation

Several non-pharmacological interventions, including acupuncture, electrostimulation, and low-level laser therapy, that administrate local stimulation to promote salivary secretion have been introduced. Acupuncture therapy involves the penetration of needles into specific points (known as acupoints) on the body surface to exert a stimulatory effect on the salivary gland. Electrostimulation utilizes electrostimulating devices to introduce electrical stimuli to the salivary reflex arch to trigger salivary secretion. Low-level laser therapy is based on the application of a laser beam to irradiate the salivary glands of patients with xerostomia. These methods are generally safe, with minimal side effects [193,194,195,196,197]. While these approaches appear to enhance salivary flow rate and alleviate subjective symptoms of dry mouth in patients with xerostomia [193,195,196,197,198,199,200,201,202], their efficacy is still controversial.

### 5.3. Gene Therapy

Gene therapy is intended to restore the lost function of salivary glands. It works by delivering genes to rectify the defects of salivary glands, eventually bringing back the normal secretory activity [203]. At present, the development of salivary gland gene therapy is still underway. Aquaporin gene therapy, which introduces recombinant adenovirus encoding aquaporin proteins into the salivary glands, illustrated a restoration of salivary movement to relieve the dry mouth phenotype in animal models and clinical trials [204,205,206]. The gene transfer of Tousled-like kinase 1B (TLK1B), a protein linked to DNA replication and DNA damage repair, to salivary glands showed a rescue of irradiation damage of the salivary gland, ameliorating hyposalivation [207,208,209]. The gene delivery of Keratinocyte growth factor (KGF), an epithelial cell-specific growth and differentiation factor, allows a regeneration of the acinar cells of salivary glands after irradiation [210,211], representing another potential treatment for xerostomia.

**Table 4 ijms-25-12815-t004:** Treatment for xerostomia.

Approaches	Principles and Effects	Advantages	Disadvantages/Limitations
Hydration (e.g., increase water consumption)	Maintaining moisture in the oral cavity [173,180,212,213,214,215]	Simple No side effects	Temporary relief of dry mouth symptoms [181,216]
Salivary stimulants (sialagogues)			
Sugar-free chewing gum	Mechanical and gustatory stimulation of salivary secretion [217]	Low cost Easily accessible Safe and fewer side effects	Short-term relief of dry mouth sensation
Malic and critic acids	Gustatory stimulation of salivary secretion [217,218]	Low cost	Short-term relief of dry mouth symptoms Lowering the pH of saliva [217,219] Demineralizing effect on the tooth enamels [217,219]
Parasympathomimetic drugs (e.g., pilocarpine and cevimeline)	Activation of the parasympathetic nervous system to promote salivary production [172,174,175,176,177]	High efficacy and prolonged effects	Significant side effects [173,178,179] Contraindicated in patients with other comorbidities [173]
Saliva substitutes	Moistening and lubrication of the oral cavity	Adequate moistening and lubricating effects Over-the-counter availability	Does not completely replicate all properties (e.g., antimicrobial property) of natural salvia Short-term relief of dry mouth symptoms Repetitive use is required
Acupuncture	Puncturing the skin at the target points of the body to provoke salivation output by stimulating the autonomic nervous system [220,221], promoting neuropeptide production [220,221], increasing blood supply to the parotid gland [221,222], and mediating tissue regeneration in radiotherapy-damaged glands [223]	Inexpensive Low risk [193,194,224,225] Positive effect persists for months after treatment [194,199,226,227]	Not applicable to individuals with irreversibly impaired salivary function Contradicted for patients having other comorbidities such as severe clotting dysfunction [223] Side effect of tiredness and tiny hemorrhage Response varies among individuals
Electrostimulation	Use of electrostimulation, either intraorally or transcutaneously, to stimulate the salivary reflex arch	Mild or no adverse events [195,196,197]	Not applicable to individuals with irreversibly impaired salivary function Response varies among individuals
Low-level laser therapy	Use of laser irradiation to stimulate salivary secretion [201,202,228,229]	Low risk	Not applicable to individuals with irreversibly impaired salivary function Response varies among individuals
Gene therapy	Delivery of gene directly to the salivary gland to restore normal salivary function [203]	Root cause treatment	Potential risks of allergic reactions and off-target effects

## 6. Research Limitations and Future Directions

The current knowledge on the oral microbial ecosystem in age-related xerostomia is scarce and indefinite. Relevant studies have demonstrated apparent alterations in oral microbial profiles in age-related xerostomia, but may encounter limitations such as unclear definition of xerostomia and small sample sizes. Therefore, explicit definitions and characterizations of dry mouth conditions, as well as an expansion in the sample size in the examinations, may give more comprehensible information on the microbial changes in age-related xerostomia. Additionally, variations in sampling and measurement methodologies, case definitions, and characteristics (e.g., primary cause of xerostomia and lifestyle) of the subjects may inevitably result in data discrepancies across various investigations. These may complicate the generalization of the oral microbial imbalance in xerostomia in the elderly population from the results of these studies. Furthermore, reciprocal influences between dysbiosis and xerostomia in older adults still remain elusive and await further detailed investigations.

Existing treatments for xerostomia have mainly focused on symptomatic relief and/or improvement in saliva secretion. These conventional strategies, such as the applications of salivary stimulants and saliva substitutes, often provide temporary relief of dry mouth symptoms, and, in some circumstances, are not tolerated by individuals experiencing xerostomia, hampering their usage in xerostomia treatment. While parasympathomimetic agents, including pilocarpine and cevimeline, are currently the most effective pharmacological stimulants in managing xerostomia, their applications are often associated with severe side effects and contraindicated in patients with comorbidities, including uncontrolled asthma, acute iritis, and narrow-closure glaucoma. Therefore, they should be used with caution, particularly in older people and patients prescribed beta-adrenergic antagonists [178], and should not be administrated in intolerant patients. In addition, the efficacy of some non-pharmacological approaches, such as salivary stimulation by acupuncture, electrostimulation, and low-level laser therapy, remains ambiguous, as responses toward these treatments may greatly vary among individuals. On the contrary, gene therapy appears to be a promising strategy to treat xerostomia, as it aims at fixing salivary gland dysfunction. Nevertheless, its implementation may be challenged by its potential risk of adverse events, such as unwanted immune system reactions and off-target effects. Currently, gene therapy is at an early stage of development; a great deal of work is required to bring it to clinical use.

Respective of the correlation between xerostomia and oral dysbiosis, restoring the oral microbial balance in patients with dry mouth symptoms may provide a therapeutic opportunity to manage xerostomia. As revealed by the clinical findings, an increased abundance of pathogenic microbes, such as *Porphyromonas*, *Fusobacterium*, *Streptococcus mutans*, and *Candidia albicans*, as well as a reduced level of opportunistic bacteria, such as *Haemophilus parainfluenza*, have been found in the elderly with xerostomia. These specific genus and species of microorganisms are the potential therapeutic targets in the treatment for xerostomia and its associated complications. At present, antimicrobial agents, such as lysozyme, lactoferrin, and lactoperoxidase, are sometimes formulated in saliva substitutes to enable protection against oral infections. Their effectiveness in antimicrobial activity is proven in in vitro study but remains controversial in clinical testing. Therefore, more efforts are required to develop effective antimicrobial additives for controlling the growth of cariogenic microorganisms in the oral cavity. Apart from the incorporation of antimicrobial agents into saliva substitutes, the use of prebiotics, probiotics, synbiotics (combinations of prebiotics and probiotics), and postbiotics may be considered as prevention and management strategies for xerostomia. These substances can help rebalance the oral microbiota by introducing beneficial microbials or inactivating the pathogenic ones [230]. They are generally cost-effective, easily available, safe, and well-tolerated, and enable long-lasting beneficial effects [231], thus demonstrating high therapeutic potential in xerostomia treatment. Meanwhile, practical considerations relating to formulation, quality controls, and potential health risks (e.g., hypersensitivity) have to be addressed to ensure effective and safe treatment with these agents. Furthermore, regarding the individual variability in microbiome profile and response to xerostomia treatment, personalized medicine based on the oral microbiota represents a potential avenue in managing dry mouth. We propose that in such a microbiota-targeted approach, the oral microbiota in a xerostomia patient can first be assessed to identify the specific microorganisms involved in dysbiosis, and then modulated by optimal treatment regimes with prebiotics and probiotics to restore its balance. Additionally, taking advantage of multi-omics analysis of the microbiome, oral microbiome profiling in patients may advance the development of effective diagnosis and precision treatments for xerostomia in the future. Overall, manipulating the oral microbiota may represent a new direction to manage xerostomia and its related complications.

## Figures and Tables

**Figure 1 ijms-25-12815-f001:**
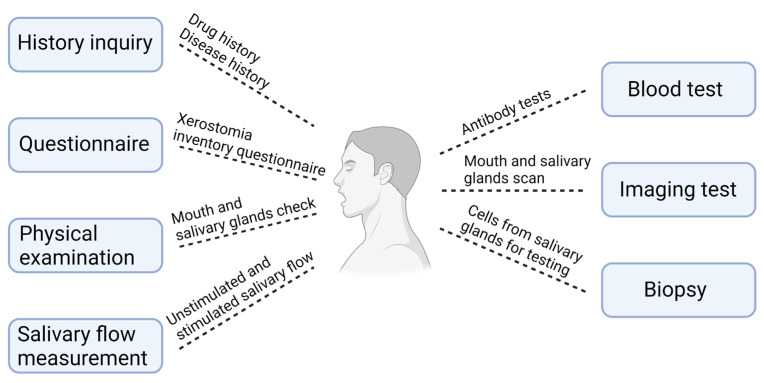
Diagnosis of xerostomia. Seven main approaches, including history inquiry, questionnaire, physical examination, salivary flow measurement, blood test, imaging test, and biopsy, are applied for the dry mouth diagnosis. Created in BioRender. Wong, C. (2024) https://BioRender.com/k08t388 (accessed on 16 October 2024).

**Figure 2 ijms-25-12815-f002:**
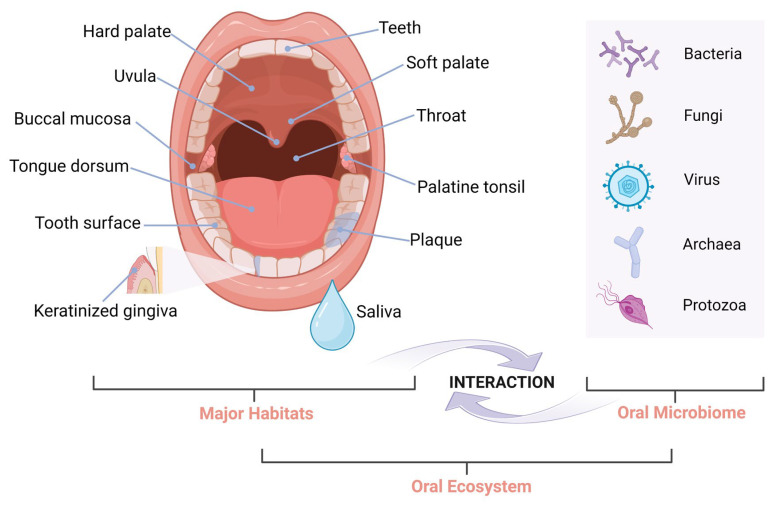
Oral ecosystem. It comprises the oral microbiome (including bacteria, fungi, viruses, archaea, and protozoa), anatomic niches in the oral cavity (the habitats where the microorganisms grow and colonize), and saliva. Interactions take place in between the oral microbiome and its habitats in the ecosystem. Created in BioRender. Wong, C. (2024) https://BioRender.com/m22z208 (accessed on 16 October 2024).

**Figure 3 ijms-25-12815-f003:**
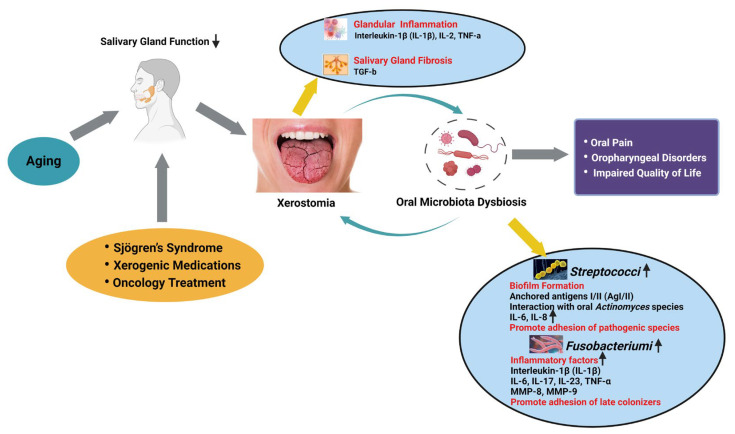
Oral microbiota dysbiosis in age-related xerostomia. The primary cause of xerostomia is attributed to the decline in salivary gland function, typically induced by aging, autoimmune disorders, the effects of xerogenic medications, or oncology treatment. These conditions can trigger severe inflammation and fibrosis within the salivary glands, leading to a reduction in saliva production. The consequent decrease in salivary flow disrupts the balance of the oral microbiome, leading to an increase in pathogenic bacteria, which further exacerbates the local inflammatory response and creates a vicious cycle. This multi-factorial interplay is ultimately responsible for the persistent symptoms of xerostomia and impairs the life quality. Created in BioRender. Wong, C. (2024) https://BioRender.com/h32o507 (accessed on 18 October 2024).

**Table 2 ijms-25-12815-t002:** Common bacteria in the oral cavity.

Type	Species	Nomenclator	Effect
Beneficial bacteria	*Lactobacillus crispatus* *	Brygoo and Aladame [74]	Antibacterial activity [75]
	*Streptococcus salivarius*	Andrewes and Horder [76]	Antimicrobial and antibiofilm activity [77,78,79]
Pathogenic bacteria	*Aggregatibacter actinomycetemcomitans*	Nørskov-Lauritsen and Kilian [80]	Cause periodontal disease [81]
	*Porphyromonas gingivalis*	Shah and Collins [82]	Cause periodontal disease [83]
	*Streptococcus mutans*	Clarke [84]	Cause caries [85]
Opportunistic bacteria	*Abiotrophia defectiva*	Kawamura et al. [86]	Cause caries [87,88]
	*Bifidobacterium dentium*	Scardovi and Crociani [64]	Cause caries [89]
	*Lactobacillus acidophilus* *	Davis [90]	Inhibit the carcinogenicity of oral streptococci [91,92] Cause caries [91,92]

* Refer to transient bacteria.

**Table 3 ijms-25-12815-t003:** Oral microbiota profiles in age-related xerostomia.

Cause of Xerostomia	Mean Age/N of Patients	Microbiota Profiles	Samples	Reference
Sjögren’s syndrome (SS)	35.5/N = 10 (control group) 58/N = 19 (xerostomia group)	The main phylum difference was *Actinobacteria*, *Firmicutes*, *Fusobacteria*, and *Proteobacteria.*	Gingival plaques	[146]
	59/N = 16 (control group) 58/N = 8 (xerostomia group)	Among low-grade xerostomia patients, salivary abundance of *H. parainfluenzae* decreased in pSS patients compared to that in non-pSS sicca patients.	Whole saliva	[23]
	47.4/N = 17 (control group) 52.4/N = 48 (xerostomia group)	Increased bacteria of the genera *Prevotella*, *Streptococcus*, *Veillonella*, *Fusobacterium*, and *Leptotrichia* and fewer bacteria of the genus *Selenomonas.*	Crevicular fluid	[147]
	53/N = 15 (control group) 56/N = 15 (xerostomia group)	At species level, *Streptococcus intermedius*, *Prevotella intermedia*, *Fusobacterium nucleatum* subsp. *vincentii*, *Porphyromonas endodontalis*, *Prevotella nancensis*, *Tannerella* spp., and *Treponema* spp. were detected in the samples from SS.	Whole saliva	[14]
	59/N = 10 (control group) 38/N = 11 (xerostomia group)	An increase in highly abundant *Streptococcus*, in addition to a decrease in *Leptotrichia and Fusobacterium* in SS.	Dorsal tongue swab	[144]
	56/N = 20 (control group) 56/N = 20 (xerostomia group)	Higher numbers and frequencies of *Streptococcus mutans*, *Lactobacillus* spp., and *Candida albicans* in the supragingival plaque; increased frequency of *C. albicans*, *Staphylococcus aureus*, *enterics*, and *enterococci. C. albicans* was detected about twice as frequently in the supragingival plaque; slightly lower proportions of *Fusobacterium nucleatum* and *Prevotella intermedia*/*Prevotella nigrescens* in the gingival crevice region.	Samples from the dorsum of the tongue, smooth mucosa, supragingival tooth surfaces, and the gingival crevice region	[148]
	54/N = 53 (control group) 57/N = 57 (xerostomia group)	*V. parvula* was significantly different compared to controls.	Subgingival plaque	[149]
Differentiated thyroid carcinoma patients (DTC) after radioiodine therapy	43/N = 32 (DTC patients without xerostomia) 40/N = 40 (healthy people) 43/N = 30 (DTC patients with xerostomia)	The abundances of *Porphyromonas*, *Fusobacterium*, and *Treponema_2* were significantly higher in DTC patients with xerostomia; *Porphyromonas* genus might be a key driver during the process of xerostomia.	Whole saliva	[145]
Type 2 diabetes	65/N = 50 (non-diabetic, 36% xerostomia) 63/N = 154 (diabetic, 62% xerostomia)	Patients with hyposalivation had significantly higher numbers of *Streptococcus mutans*, *Lactobacillus* spp., and *Candida* spp. in the saliva compared with those without hyposalivation.	Whole saliva	[150]
Antihypertensive medications	35–76/N = 200 (control group, 25.5% xerostomia) 44–89/N = 200 (ambulatory hypertensive patients taking antihypertensive medications, 50% xerostomia)	The mean levels of *Streptococcus mutans*, *Lactobacilli* spp., and *Candida* spp. in the medicated hypertensive group were significantly higher than in the control group.	Whole saliva	[151]
Radiotherapy (RT) for head and neck cancer patients	52.4/N = 28 (control group) 56.2/N = 28 (RT treatment group, xerostomia)	*Candida* and family *Enterobacteriaceae* showed increased prevalence with RT, and were associated with the occurrence of mucositis and xerostomia.	Supra and subgingival biofilms	[152]
	26–70/N = 8 (head-and-neck cancer patients)	Higher doses of radiation lead to a stronger species reduction; before radiation therapy, the most abundant phyla were, in order of prevalence, *Proteobacteria*, *Firmicutes*, *Bacteroidetes*, and *Actinobacteria*; during radiation therapy, the order was changed to *Firmicutes*, *Actinobacteria*, *Proteobacteria*, and *Bacteroidetes.*	Before and during RT, a pooled supragingival plaque sample from the buccogingival surfaces of the maxillary first molar were collected at 7-day intervals	[153]
	53/N = 13 (control group) 53/N = 13 (RT treatment group, xerostomia)	*Candida albicans* was found in one or more sites in 54% of the RT subjects and in 15% of the controls; in supragingival plaque, *Lactobacillus* spp. was detected in 92% of the RT subjects, and the number and proportion of *Lactobacillus* spp. were extremely high compared with the controls.	Tongue, buccal mucosa, vestibulum, supragingival plaque, and subgingival region	[154]

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
