# Peer review of "The Oral Microbial Ecosystem in Age-Related Xerostomia: A Critical Review"

_ijms, 2024, doi:10.3390/ijms252312815_

Round 1

Reviewer 1 Report

Comments and Suggestions for Authors

The article addresses a topic of great clinical relevance, with potential for significant impact on applied practice. The information presented throughout the text is well-organized and covers essential aspects that capture the reader's interest and attention. However, although the work is a literature review, it is crucial to include a detailed description of the methodology employed in the study. It is recommended to present key points, such as the search strategy used, specifying the databases consulted.

It is suggested to include a flowchart illustrating the article selection process, detailing the number of studies found in each database, the inclusion and exclusion criteria adopted, and whether any software was used to assist in the selection process. Additionally, it is important to specify the languages and types of studies included, as well as the time frame of the publications analyzed.

Given the high number of references (over 200), I suggest adopting a more current and focused approach to avoid a superficial and overly generalized analysis, ensuring that the most recent and relevant studies are thoroughly discussed. This would contribute to greater consistency and relevance in the conclusions drawn.

The use of images in the article is excellent and adds significant value to the content, sparking greater interest in the topic. However, the importance of refining the methodology is reinforced to ensure the scientific quality and clarity of the review presented. It is recommended to include a more detailed critical analysis of the studies mentioned, allowing the reader to clearly identify the gaps in the literature and opportunities for future research. In particular, it would be valuable to discuss the potential biases of the studies and the limitations of the therapeutic interventions addressed.

The “future directions” section presents promising proposals, such as gene therapy, but could be expanded with a more in-depth discussion of the clinical implications of the findings, as well as the challenges in implementing some of these approaches, such as probiotic therapy and genetic interventions. Exploring the potential for personalized treatments based on the oral microbiota could further enrich this discussion.

In conclusion, the article has merit and great potential but requires methodological adjustments to consolidate its scientific contribution.

Reviewer 2 Report

Comments and Suggestions for Authors

ijms-3299292

The manuscript “The Oral Microbial Ecosystem in Age-Related Xerostomia: A Critical Review (ijms-3299292)” covers a broad range of information on age-related xerostomia. However, genetic and molecular mechanisms for age-related xerostomia are still not explained in detail. Overall, the manuscript is convincing and has some points to be considered by the authors.

1.    Table 1: Table 1 shows the causes of xerostomia. Is this xerostomia caused by age or showing overall xerostomia? I suggest focusing more on age-related changes and sticking with the title of the manuscript.

2.    Table 1: The references for Symptomatic xerostomia are missing. Please add the appropriate references. In addition, the last point from True Xerostomia and Symptomatic Xerostomia is confusing. Both groups have covered lifestyle (coffee/tea intake) as the cause of xerostomia.

3.    In Section 3.1, Oral Ecosystem and Oral Microbiomes: The authors discussed more about the general oral ecosystem and microbiomes, but the correlation of microflora with age-related xerostomia is severely lacking. Please explain more about the relation between age-xerostomia and oral microflora.

4.    Table 2: The habitat of the species is extended to the Urogenital or Vagina which is beyond the scope of the manuscript. The author should focus only on the oral cavity.

5.    All the figures are not explained in the figure legends. Please explain.

6.    Table 3 shows the mean age is not much higher in all the conditions. In particular, SS have a similar age range in the control group and xerostomia group. How can authors explain this? 

7.    Table 3: It will be more promising if the authors differentiate the microbiota profiles from the control group and xerostomia separately rather than combining them in the same category.

Round 2

Reviewer 1 Report

Comments and Suggestions for Authors

Dear authors

We appreciate the feedback provided and believe these modifications significantly improve the quality and clarity of the manuscript.